# Validation of an omega-3 substrate challenge absorption test as an indicator of global fat lipolysis

Steven D. Freedman[1], Kamil Zaworski[2], Kateryna Pierzynowska[2,3,4], Stefan Pierzynowski[3,5,6], Robert Gallotto[4], Meghana Sathe[7], Drucy S. Borowitz[8]*

1 Division of Gastroenterology, Beth Israel Deaconess Medical Center, Boston, MA, United States of America, 2 Department of Animal Physiology, The Kielanowski Institute of Animal Physiology and Nutrition, Polish Academy of Sciences, Jabłonna, Poland, 3 Department of Biology, Lund University, Lund, Sweden, 4 Synspira Therapeutics, Inc., Framingham, MA, United States of America, 5 Anara AB, Trelleborg, Sweden, 6 Department of Medical Biology, Institute of Rural Health, Lublin, Poland, 7 Division of Pediatric Gastroenterology, Hepatology and Nutrition, University of Texas Southwestern/Children's Health, Dallas, TX, United States of America, 8 Department of Pediatrics, Jacobs School of Medicine and Biomedical Sciences, University at Buffalo, Buffalo, NY, United States of America

* borowitz@buffalo.edu

## Abstract

### Introduction

The coefficient of fat absorption (CFA) quantifies fat that remains in stool after digestion and is not a direct measure of lipolysis. CFA has been used to assess treatment of pancreatic insufficiency but does not correlate with pancreatic enzyme replacement therapy dose. We explored use of an omega-3 substrate absorption challenge test as a sensitive test of lipolysis and absorption.

### Methods

We studied a novel microbially-derived lipase (SNSP003) employing an established surgical model commonly used to study the uptake of macronutrients, the exocrine pancreatic insufficient pig. Pigs were fed a high-fat diet and given a standardized omega-3 substrate challenge to test the effect of lipolysis on its absorption. Blood was drawn at 0, 1, 2, 4, 6, 8, 12, and 24 hours following the substrate challenge and was analyzed for omega-3 and total fat levels (c14:c24). SNSP003 was also compard to porcine pancrelipase.

### Results

The absorption of omega-3 fats was significantly increased following administration of 40, 80 and 120 mg SNSP003 lipase by 51% (p = 0.02), 89%, (p = 0.001) and 64% (p = 0.01), respectively, compared to that observed when no lipase was administered to the pigs, with $T_{max}$ at 4 hours. The two highest SNSP003 doses were compared to porcine pancrelipase and no significant differences were observed. Both doses increased plasma total fatty acids (141% for the 80 mg dose (p = 0.001) and 133% for the 120 mg dose (p = 0.006), compared

experiment are available at doi: 10.5114/aoms.2018.73471 and for the DHA analysis at https://www.ncbi.nlm.nih.gov/pubmed/22623386.

**Funding:** Synspira Therapeutics provided support in the form of salaries for SF, MS and DB. The specific roles of these authors are articulated in the 'author contributions' section. The funders had no role in study design, data collection and analysis, decision to publish, or preparation of the manuscript.

**Competing interests:** The research was sponsored by Synspira Therapeutics, Inc. SF, MS and DB are members of the Synspira Therapeutics Scientific Advisory Board and are reimbursed for their efforts. DB is a member of the Synspira Therapeutics Board of Directors and receives compensation in stock. RG is the President and CEO of Synspira Therapeutics, Inc. There are no patents, products in development or marketed products associated with this research to declare. The product tested (SNSP003) is about to enter human clinical trials and thus is in development, and patents have been filed on SNSP003. This does not alter our adherence to PLOS ONE policies on sharing data and materials.

to no lipase) and no significant differences were observed between the SNSP003 lipase doses and porcine pancrelipase.

## Conclusion

The omega-3 substrate absorption challenge test differentiates among different doses of a novel microbially-derived lipase and correlates with global fat lipolysis and absorption in exocrine pancreatic insufficient pigs. No significant differences were observed between the two highest novel lipase doses and porcine pancrelipase. Studies in humans should be designed to support the evidence presented here that suggests the omega-3 substrate absorption challenge test has advantages over the coefficient of fat absorption test to study lipase activity.

## Introduction

The coefficient of fat absorption (CFA) is a test that was developed in the 1950's and has been used for two purposes: to diagnose exocrine pancreatic insufficiency (EPI) and to determine performance of pancreatic enzyme replacement therapy (PERT). CFA quantifies fat that remains in stool after digestion (lipolysis) and absorption and thus the results are affected by all causes of fat malabsorption, not just pancreatic insufficiency [1]. The current commercially available PERTs came to the market prior to the creation of the US Food and Drug Administration (FDA). Because of their critical role in preventing life-threatening malnutrition in people with EPI, such as those with cystic fibrosis (CF), they were initially available for clinical use without having undergone the usual studies to determine safety and dose. Then in response to an epidemic of fibrosing colonopathy, an adverse event attributed to PERT, these medications had to prove safety in manufacturing and efficacy compared to placebo. Between 2009 and 2012, six manufacturers submitted efficacy data to the US FDA, using CFA as the endpoint. However, there is no correlation between the CFA measurement and PERT dose [2–4]. As stated by the Gastrointestinal Drugs Advisory Committee during the FDA PERT approval process, there is no consensus on what would constitute a clinically meaningful change in CFA [5]. Thus, it is not an optimal test for PERT drug development, although it can distinguish treatment from placebo in people with EPI. A better outcome is needed to advance treatment and support clinical management for people with EPI. Ideally, this outcome would focus on quantitating the functional action of lipase, one of the critical enzymes absent in people with fat malabsorption.

Dietary triglycerides, including long-chain polyunsaturated fats (LCPUFA) must be hydrolyzed by pancreatic lipase into free fatty acids and monoglycerides before they can be absorbed by enterocytes in the small intestine. LCPUFAs include the omega-3 fats docosahexaenoic acid (DHA; 22:6n-3) and eicosapentaenoic acod (EPA; 20:5n-3). The conversion of essential fatty acids in the enterocytes such as linoleic acid to DHA is negligible, and there is minimal conversion to EPA [6–9]. Therefore, since there is a low rate of synthesis in the body and the primary source of DHA and EPA is from the diet, plasma and tissue levels of these fats reflect lipolysis and absorption of this fat from exogenous sources. DHA and EPA are among the most challenging fatty acids to digest and absorb due to the length of their carbon chains and multiple double bonds [10]. If levels of these omega-3 fats increase, it is reasonable to assume that lipolysis and absorption of all shorter chain and more saturated fatty acids would increase as well.

Thus, measurement of plasma levels of DHA and EPA following ingestion of a standardized dose could be used as a direct indicator of lipase activity and global fat absorption.

We sought to study an omega-3 substrate absorption challenge test (SACT) in an EPI pig model to demonstrate that: 1) the omega-3 SACT could effectively differentiate the degree of lipolysis and absorption among escalating doses of SNSP003, a novel, microbially-derived lipase; 2) that the omega-3 SACT could detect the lipolytic activity of commercially-available porcine PERT and 3) that lipolysis and absorption of the full range of dietary fats correlates with the omega-3 SACT. In addition, we aimed to determine the time to maximal absorption (Tmax), concentration peak (Cmax), and area under the curve (AUC) at time points that would help to define physiological effects of lipolysis and absorption.

## Materials and methods

This study with an adaptive design followed principles of Good Laboratory Practice as defined by the Organization for Economic Co-operation and Development as applicable. Procedures involving the care and the use of animals in this study were reviewed and approved by the Second Warsaw Local Ethics Committee for Animal Experimentation in Warsaw, Poland, permission number WAW2/15/2017. During the study, the care and use of animals was conducted in accordance with the principles outlined in the current Guide to the Care and Use of Experimental Animals. Details concerning the ethical treatment of animals, including description of anesthesia, analgesia and euthanasia, are in the supplementary materials.

### Exocrine pancreatic insufficient (EPI) pig model

The EPI pig is an established surgical model commonly used to study the uptake of macronutrients and to evaluate different preparations of orally administered pancreatic enzymes [11–13].

EPI pigs were individually housed in collection cages equipped with a dry feeding trough, a drinking nipple, and a constant heating lamp. All pigs were allowed to move freely within their cages and had visual contact with one another and were weighed once a week. During the study period, pigs were fed an amount equivalent to 4% of their body weight daily with 1% given at the morning meal and approximately 3% at the afternoon meal. Meals were solid pig chow containing 3% fat, 21% protein, 73% carbohydrates and 3% minerals/vitamins (Morawski Plant, Kcynia, Poland). Following the surgery, and beginning on Day -7, they were fed a high fat diet containing 20% fat, 17.5% protein, 57.3% carbohydrates and 5.2% minerals/vitamins (HFD 20, Kcynia, Morawski Plant, Poland). The morning meal of the SACT mimicked a standard high-fat meal for a person with CF (fat 36–40%, protein 15–20%, carbohydrate 40–49%).

The experiment was conducted on 12 EPI pigs that weighed ~13.5 kg (range 10–16 kg). The sample size was based on previous experiments using the EPI pig model and was estimated using G*Power software, version 3.1.9.7 for ANOVA with repeated measurements and within-between interactions, at $\alpha = 0{:}05$ with 95% power, assuming $f$ (effect size) = 1.2, for five study groups and five measurements, taking into consideration the correction for repeated measurements (0.5). The calculation yielded the total sample size of 10 animals and the 12 animals were taken considering the possible risk of EPI surgery failure [12–14]. Each pig had eight 3–5 mL blood samples drawn over the 24-hour SACT period at the following time points: 0, 1, 2, 4, 6, 8, 12, and 24 hours, beginning at 8 AM. Blood samples were taken from the jugular vein catheter to 5 ml syringes (BD Syringe™, SKU 309646, BD Switzerland Sarl, 1262 Eysins, Vaud. Switzerland) and then transferred to BD Vacutainer® EDTA Tubes (SKU 366643, BD Switzerland Sarl, 1262 Eysins, Vaud. Switzerland). The collected blood was centrifuged at 4,000 rpm at 4˚C for 10 min. Plasma was separated and stored at -80C. Plasma was analyzed within

one month after study completion. Pigs were divided into two groups with block randomization based on their degree of steatorrhea and body weight. These groups only differed in the sequence of administration of the various enzyme doses to minimize possible period effects.

## Fatty acid analysis

Fatty acid analysis was carried out using a GC2010 Gas Chromatopgraph (Shimadzu) equipped with a SP2560 100-m fused silica capillary column [15]. Plasma was transferred to a screw-cap glass vial and BTM (methanol containing 14% boron trifluoride, toluene, methanol; 35:30:35 v/v/v) (Sigma-Aldrich, St. Louis, MO) was added. The vial was briefly vortexed and heated in a hot bath at 100˚C for 45 minutes. After cooling, hexane (EMD Chemicals, USA) and HPLC grade water was added, the tubes were recapped, vortexed and centrifuged help to separate layers.

Fatty acids were identified by comparison with a standard mixture of fatty acids (GLC 782, NuCheck Prep, Elysian, MN) which was also used to determine individual fatty acid calibration curves. The following 24 fatty acids (by class) were identified: saturated (14:0, 16:0, 18:0, 20:0, 22:0 24:0); *cis* monounsaturated (16:1, 18:1, 20:1, 24:1); trans [16:1, 18:1, 18:2,); *cis* n-6 polyunsaturated (18:2, 18:3, 20:2, 20:3, 20:4, 22:4, 22:5); *cis* n-3 polyunsaturated (18:3, 20:5, 22:5, 22:6). Fatty acid composition was expressed as a percent of total identified fatty acids.

## Test articles

A recombinant I.2 class lipase variant was developed through molecular engineering of the wild type to be stable under physiologically relevant low pH conditions and resistant to proteolysis in the fed-state stomach and small intestine. Lipase stability was accomplished by making only a few changes in the amino acid sequence outside of the active site and calcium binding sites, well within the natural variation of these lipase classes. This novel lipase (SNSP003) is formulated with protease and amylase (only results of lipase testing are presented here). SNSP003 was given at lipase doses of 20 mg, 40 mg, 80 mg, and 120 mg.

Commercially available porcine-derived pancrelipase (Kreon® 25 000, Mylan Healthcare, Warsaw, Poland) was provided as 50,000 lipase units per meal, at a per kg dose ~50% higher than maximum recommended dose for people with CF and EPI.

DHA and EPA were provided during each SACT as commercially available 2,000 mg Omega-3 softgels (Kinoko Life, Spain), which contain 500 mg EPA and 250 mg DHA per capsule. Six whole capsules were given to pigs with the first portion of feed making the total dose for the SACT 3000 mg EPA and 1500 mg DHA.

## Statistical analysis

Statistical analysis was performed on the data generated from this study using the Brown-Forsythe and Welch ANOVA when evaluating multiple comparisons for normally distributed datasets or Kruskall-Wallis test with uncorrected Dunn's test for multiple comparisons when data was not normally distributed (GraphPad Prism 8.4, USA). Differences were considered significant if $p \leq 0.05$; differences were considered as a trend when $p \leq 0.1$; data with Gaussian distribution are expressed as mean ± standard deviation (± SD), data with non-Gaussian distribution are expressed as median ± intraquartile range (± IQR). The data was tested for normal distribution using the Shapiro-Wilk normality test. Outliers within data sets were identified using the ROUT method of regression, using (Q = 0.5%).

## Results

Raw data for the Results presented below are available in the online supplement.

### Effect of SNSP003 on DHA and EPA as products of lipolysis

The effect of the different doses of SNSP003 lipase on DHA and EPA absorption is shown in Fig 2A. The absorption of DHA and EPA was significantly increased following administration of the 40, 80 and 120 mg lipase dose by 51% (p = 0.02), 89%, (p = 0.001) and 64% (p = 0.01), respectively, compared to that observed when no lipase (NL) was administered to the pigs, with $T_{max}$ at 4 hours (Fig 1A). The calculated $AUC_{24}$ for both the 80 mg and the 120 mg lipase dose resulted in a significant increase in the mean plasma DHA and EPA concentration by 83% (p = 0.001) and 62% (p = 0.01), respectively when compared to when no lipase was administered. (Fig 1B). Fig 1C shows the $C_{max}$ for plasma DHA and EPA, following administration of the different lipase doses. A significant increase in $C_{max}$ of 60%, compared to the no lipase administration, was observed following administration of the 80 mg lipase dose (p = 0.009). A similar but slightly lower increase in $C_{max}$ was seen with the 120 mg dose increasing $C_{max}$ by 45% (p = 0.05) and the 40 mg dose by 40%, (p = 0.07).

### Lipolytic effects of the most efficacious SNSP003 doses compared to porcine pancrelipase on DHA and EPA

Based on the data above, the 80 mg and 120 mg SNSP003 lipase doses were found to be the most efficacious in EPI pigs, thus the plasma uptake of fatty acids following these SNSP003 lipase doses were compared to plasma levels when the SACT was performed using porcine pancrelipase. Pigs received a maximum dose of 600 mg of pancrelipase equivalent to approximately 3,750 lipase units/kg/meal (range 2,750–4,900).

Both the SNSP003 doses and porcine pancrelipase resulted in a significant uptake of DHA and EPA over 24-hours (Fig 2A). Plasma DHA and EPA $AUC_{24}$ was increased by 126% (p = 0.007) for the 80 mg doses, 141% for the 120 mg dose (p = 0.09) and 122% for the pancrelipase dose (p = 0.027) compared to no enzyme (Fig 2B). $C_{max}$ for plasma DHA and EPA concentration over 24-hours increased by 211% (p = 0.01) for the 80 mg doses, 264% for the SNSP003 120 mg dose (p = 0.027) and 115% for the pancrelipase dose (p = 0.001) compared to no lipase (Fig 2C). No significant differences were observed between the lipase doses and pancrelipase.

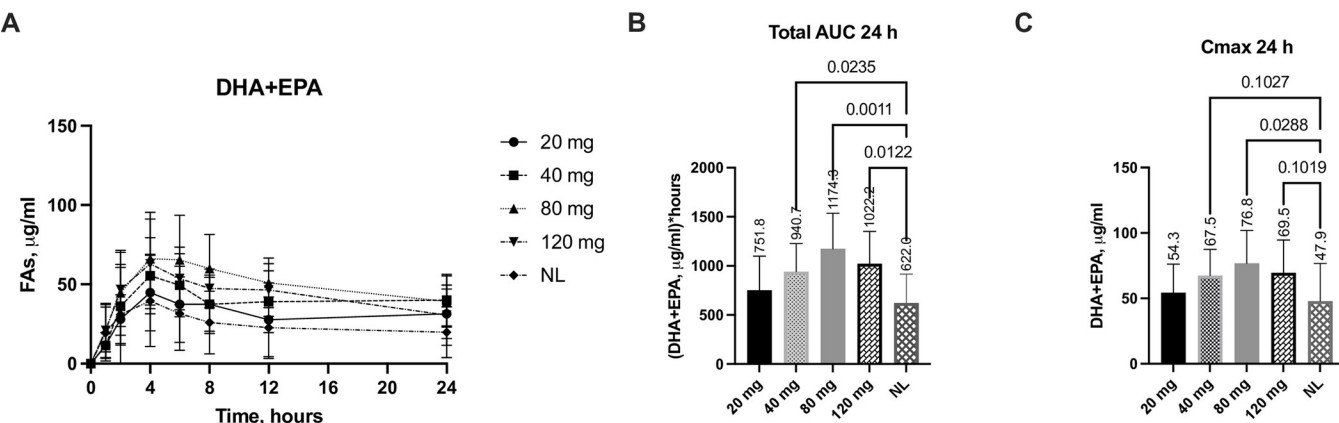

**Fig 1.** Changes from baseline in plasma DHA + EPA (A), 24-hour area under the curve (AUC) (B), and $C_{max}$ over 24 hours (C) following administration of various SNSP003 lipase doses (20 mg, 40 mg, 80 mg and 120 mg), compared to no lipase (NL). All data are baseline adjusted. Data are presented as Mean± SD.

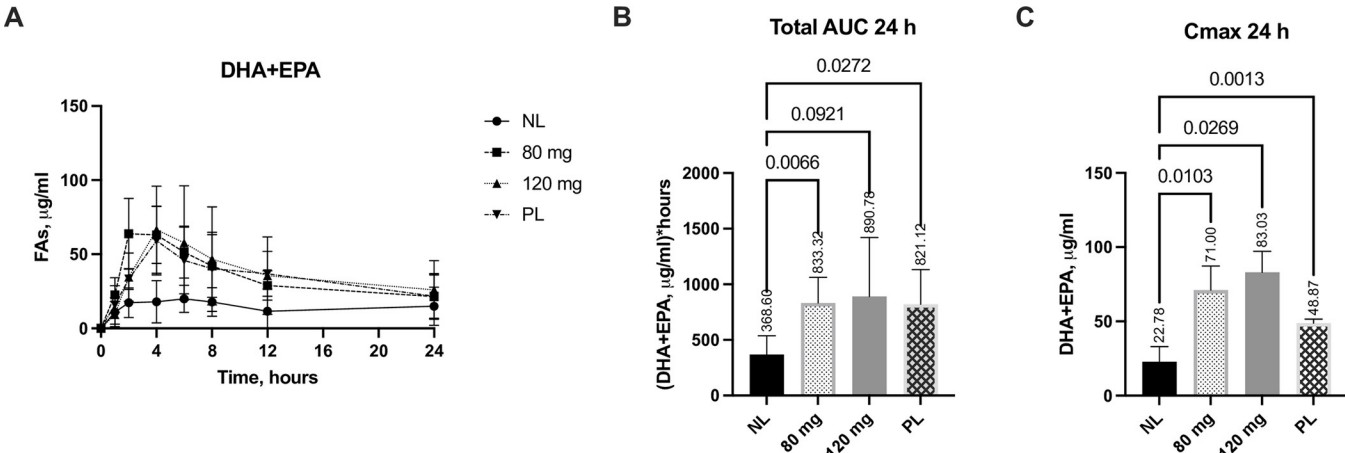

**Fig 2.** Changes from baseline in plasma concentration DHA and EPA (A), 24-hour area under the curve (AUC) (B), and Cmax over 24-hours (C), following administration of SNSP003 80 mg or 120 mg lipase doses or a 600 mg dose of pancrelipase (PL), compared to no lipase (NL). All data are baseline adjusted. All data are presented as Mean ±SD, $C_{max}$ data are presented as Median ± IQR.

## Lipolytic effect of SNSP003 on a broad spectrum of fatty acids

The effect of different doses of SNSP003 lipase on total long-chain fatty acids (n = 24; c14:c24) when given with a fixed high-fat meal and substrate challenge is shown in Fig 3A. Plasma concentration of total long-chain fatty acids peaked at 2-hours and returned to baseline eight hours after consumption of the morning SACT meal. Baseline values were subtracted when calculating area under the curve (AUC) and $C_{max}$. $AUC_8$ values for plasma total fatty acids, following administration of the various lipase doses, are shown in Fig 3B. Both the lipase doses demonstrated a considerable increase in plasma total long chain fatty acids by 141% for the 80

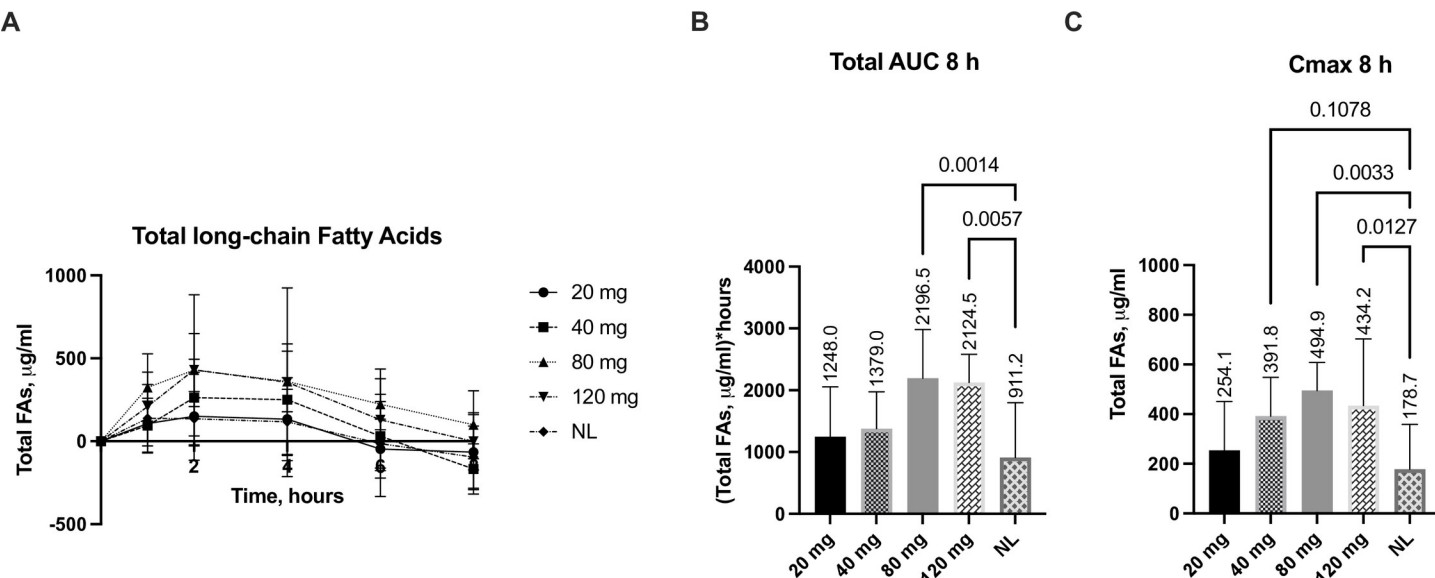

**Fig 3.** Changes from baseline in plasma total fatty acids concentration (A), 8-hour area under the curve (AUC) (B), and $C_{max}$ over 8 hours (C) following administration of various SNSP003 lipase doses (20 mg, 40 mg, 80 mg and 120 mg), compared to no lipase (NL). All data are baseline adjusted. AUC and $C_{max}$ data are presented as Median ±IQR, other data are presented as Mean ±SD.

mg dose (p = 0.001) and 133% for the 120 mg dose (p = 0.006), compared to no lipase. An increase in plasma concentration was also observed for the 40 mg and 120 mg doses. Finally, a significant positive effect of escalating doses of lipase on total fat absorption was reflected in the median $C_{max}$ values for plasma total fatty acids with both the 80 mg by 128% (p = 0.003) and the 120 mg by 157% (p = 0.013) compared to no lipase (Fig 3C).

## Lipolytic effects of the most efficacious SNSP003 doses compared to porcine pancrelipase on a broad spectrum of fatty acids

Both the SNSP003 doses and porcine pancrelipase resulted in a significant uptake of total fatty acids over 8-hours (Fig 4A). Plasma total fatty acids $AUC_{24}$ was increased by 108% (p>0.1) for the SNSP003 80 mg doses, 98% for the 120 mg dose (p>0.1) and 90% for pancrelipase (p>0.1) compared to no lipase (Fig 4B). $C_{max}$ for plasma total fatty acids concentration over 24-hours increased by 203% (p = 0.001) for the SNSP003 80 mg doses, 153% for the 120 mg dose (= 0.05) and 109% for the pancrelipase dose (p = 0.11) compared to no lipase (Fig 4C). No significant differences were observed between the SNSP003 lipase doses and pancrelipase.

## Discussion

We have demonstrated in a pig model of EPI that the omega-3 SACT can distinguish among the enzymatic activity of different doses of lipase lipase by directly assessing absorption of the by-products of fat digestion in plasma with acceptable between-subject variability. Although CFA has been the criterion standard to determine PERT efficacy, a Cochrane review of PERT for people with CF, updated in 2020, noted a wide range of CFA in the studies evaluated and in the response to PERT among patients [16]. This is consistent with observations made in 1960 by Weijers, Drion and van de Kamer that discussed the clinical application of fat balance studies. They noted that the CFA in patients with fat malabsorption had much greater variability than the CFA in a healthy population, limiting its use for dose-ranging studies [17]. Our data demonstrated the ability of the omega-3 SACT to detect differences among different lipase doses in EPI pigs. A previous study in this same pig model showed that the total fecal fat

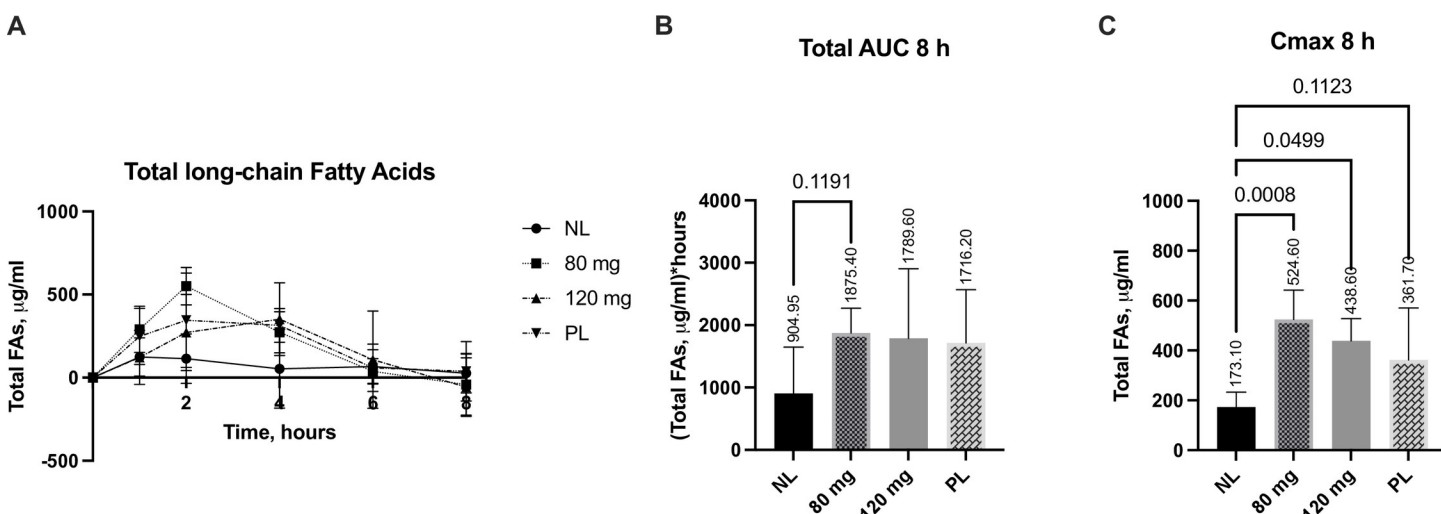

**Fig 4.** Changes from baseline in plasma total fatty acid levels over 8-hours (A), 8-hour area under the curve (AUC) (B), and $C_{max}$ over 8 hours (C), following administration of SNSP003 80 mg or 120 mg lipase doses or a 600 mg dose of pancrelipase (PL), compared to no lipase (NL). All data are baseline adjusted. $C_{max}$ data are presented as Median±IQR. Other data are presented as Mean ±SD.

content, which is the main variable in CFA calculations, is strongly correlated with plasma levels of DHA and EPA [18]. This correlation suggests that the SACT, which can quantitate dose-response, measures fat malabsorption in a manner similar to CFA, the current but less precise criterion standard. Others have demonstrated an increase in fatty acid area under the curve and $C_{max}$ after use of an in-line lipase cartridge to aid in digestion of liquid meals in people with CF who have EPI [19]. These investigators studied a fixed dose of lipase. Thus they demonstrated the utility of this measure of lipolysis in people with EPI but not its ability to distinguish among different doses.

Taken together, the variability of CFA and the ability of the omega-3 SACT to distinguish among doses of lipase supports the idea that this blood-based, pharmacokinetic-like test of lipolysis and absorption is a significant improvement over CFA to measure lipolysis and fat absorption.

In addition, we have shown that two different doses of the novel, microbially derived lipase SNSP003 demonstrate equivalent values for plasma uptake of DHA and EPA and total fatty acids (C14:C24) when compared to porcine pancrelipase. This strengthens the argument that the omega-3 SACT could be used as a replacement for CFA, since commercially available PERTs were approved using CFA as the registration endpoint. Lipolysis and absorption using a cartridge containing lipase, was studied in people with CF receiving enteral feedings using an omega-3 SACT and results were associated with a decrease in abdominal pain and bloating. [20] There is a paucity of clinical correlations with longer-term outcomes and CFA. Body mass index (BMI) is considered the main nutritional outcome measure in people with CF because it correlates with lung function and survival [21]. Fat digestion and absorption should be strongly associated with this clinically relevant outcome, however CFA does not appear to be valid in this aspect. An evidence-based review found "insufficient evidence to make a recommendation regarding the association of specific PERT dosing and CFA or growth" [21]. Although not contemporaneously studied, use of an in-line lipase cartridge, which has been shown to be associated with improvements in plasma DHA and EPA absorption relative to placebo, has been shown to lead to improved BMI and GI symptoms when used over one year [22]. Use of this in-line lipase cartridge was also associated with increases in red blood cell (RBC) membrane DHA and EPA over 90 days [19]. RBC levels of DHA and EPA, known as the omega-3 index, can be obtained as a home-based fingerstick blood test and has been used in various disease populations with normal fat absorption [23–26]. This opens the possibility of having paired tests: the plasma omega-3 SACT for short-term assessment of dose and the RBC omega-3 index to correlate with longer term fat absorption and clinical outcomes. This could be viewed as similar to use of blood glucose for acute management of insulin dose and hemoglobin A1c for longer term assessment of diabetes control.

Although DHA and EPA have many positive attributes as nutrients, clinicians want to know that the full range of fats has been absorbed. Our data support our expectation that absorption of DHA and EPA would reflect the lipolysis and absorption of the full range of dietary fats. We were able to demonstrate peak digestion and absorption of omega-3 fats at 4 hours after ingestion, and that of easier-to-digest total fats at approximately 2 hours. This feature of the omega-3 SACT enables assessment of the timing of lipase activity, which cannot be achieved using CFA. This time to peak absorption is similar to that seen in people without fat malabsorption who are given omega-3 fats as pre-digested free fatty acids [27]. Furthermore, we showed that the area under the curve was similar between the two highest doses of SNSP003 and pancrelipase, suggesting that the exposure to enzymatic activity is similar. The maximal concentration of plasma fats was greater for the two highest doses of SNSP003 doses than for pancrelipase, implying that SNSP003 lipase is efficacious in digestion and absorption of the full spectrum of ingested fats. The level of detail available when using a

pharmacokinetic-like test provides much greater information about lipase activity by directly assessing fat absorption rather than being a "black box" method such as CFA.

A limitation in this study is that the sample size is limited. However, the effect size was large, therefore even with a small sample we were able to demonstrate clear differences based on lipase dose. The EPI pig model has been used to study EPI and PERT [12], however pigs are not people. Nonetheless, we believe this data supports the use of this test in humans. Distribution constraints during the COVID pandemic and limitations to omega-3 supplements led us to use a DHA supplement that included both ethyl esters and DHA triglyceride. Ethyl esters are not metabolized via pancreatic lipase. This may have accounted for some of the long tail seen in the area under the curve, which does not come back to baseline by 24-hours, unlike the pattern seen with total fats. Going forward, pure DHA triglyceride, the omega-3 found in fish, should be the substrate used to study lipolysis. The omega-3 SACT in EPI pigs was a single dose study, thus clinical correlation was not possible. Last, it should be noted that pancrelipase is a porcine-derived product, thus it is an ideal lipase for pigs but not necessarily for humans.

## Conclusions

We have demonstrated that the omega-3 substrate absorption test can distinguish among differing doses of lipase in a pig model of exocrine pancreatic insufficiency by assessing fat absorption in plasma. Use of the EPI pig model has enabled us to develop an outcome measure that can be a significant improvement over CFA for PERT drug development and clinical mangement. The results obtained with the most efficacious doses of a novel lipase correspond to those seen with a commercially available pancrelipase. The omega-3 SACT correlates with global fat lipolysis and absorption. Studies in humans should be designed to support the evidence presented here that suggests the omega-3 SACT has advantages over CFA to study lipase activity and has the potential to be useful as a new outcome measure for clinical research.

## Supporting information

**S1 Data.**
(XLSX)

**S1 File. Ethical treatment of exocrine pancreatic insufficient pigs.**
(DOCX)

## Author Contributions

**Conceptualization:** Steven D. Freedman, Robert Gallotto.

**Formal analysis:** Kateryna Pierzynowska.

**Investigation:** Kamil Zaworski, Kateryna Pierzynowska.

**Methodology:** Kamil Zaworski, Kateryna Pierzynowska, Stefan Pierzynowski.

**Project administration:** Robert Gallotto.

**Supervision:** Kateryna Pierzynowska, Stefan Pierzynowski.

**Writing – original draft:** Steven D. Freedman, Kateryna Pierzynowska, Drucy S. Borowitz.

**Writing – review & editing:** Steven D. Freedman, Kateryna Pierzynowska, Stefan Pierzynowski, Robert Gallotto, Meghana Sathe, Drucy S. Borowitz.

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
