## [Decision Letter · Decision Letter 0]

7 Mar 2023

PONE-D-23-01728Validation of an omega-3 substrate challenge absorption test as an indicator of global fat lipolysisPLOS ONE

Dear Dr. Borowitz,

Thank you for submitting your manuscript to PLOS ONE. After careful consideration, we feel that it has merit but does not fully meet PLOS ONE’s publication criteria as it currently stands. Therefore, we invite you to submit a revised version of the manuscript that addresses the points raised during the review process.

We look forward to receiving your revised manuscript.

Kind regards,

Ishtiyaq Ahmad, Ph.D

Academic Editor

PLOS ONE

Journal Requirements:

3. In the competing interests statement within the manuscript and in the online submission form, please declare your affiliation with Synspira Therapeutics and thoroughly report any potential competing interests related to this affiliation. 

 "The sponsor, Synspira Therapeutics, funded the animal experiment and participated in study design, decision to publish, and preparation of the manuscript."  

**Additional Editor Comments:**

I totally agree with the reviewer 1 comments. Authors must adress all the queries raised by the reviewer.

These queries are geniune so make sure all queries should be answered so as to proceed it for further process.

**Reviewers' comments:**

Reviewer's Responses to Questions

**Comments to the Author**

1. Is the manuscript technically sound, and do the data support the conclusions?

Reviewer #1: Yes

2. Has the statistical analysis been performed appropriately and rigorously? 

Reviewer #1: Yes

3. Have the authors made all data underlying the findings in their manuscript fully available?

Reviewer #1: No

4. Is the manuscript presented in an intelligible fashion and written in standard English?

Reviewer #1: Yes

5. Review Comments to the Author

Reviewer #1: The manuscript is well written, technically sound and with good use of language. The introduction portion is relevant. Methodology section needs to be improved. Statistics is robust and meticulously performed. The discussion section is well written. Overall, the manuscript can be considered for further evaluation after including the minor corrections

**Comments to authors**

**Under Abstract **

The section is well written, conclusion is very much clear and concise

**Under Introduction **

Comments

Overall the introduction is good and informative. I would only suggest to put some lines about the functional role of the omega-3 substrate challenge absorption test in this section rather than focusing more on the critical aspect of the coefficient of fat absorption (CFA) test.

**Under Methodology**

The methodology employed is robust, however information about following points need to be addressed

The methodology portion should be limited to the information about use of methods. You have added the background of exocrine pancreatic insufficient (EPI) pig model which is not needed here. Give information about the following points

Is the sample size sufficient?

How many replications were performed?

Time of blood sampling?

Sampling procedure, use of syringes and anticoagulant use is missing, time gap between blood sampling and analysis.

Features of the column use for fatty acid determination in GCMS.

**Under statistical analysis  **

Robust

**Under results and discussion**

Both the results are very informative. Well written.

No major changes need in this section. You should further relate your study with other such work being carried over recently.

Try to make this portion more coherent and improve the figure quality.

6. PLOS authors have the option to publish the peer review history of their article (what does this mean?). If published, this will include your full peer review and any attached files.

Reviewer #1: **Yes: **Naveed Nabi

---

## [Author Response · Author response to Decision Letter 0]

23 Mar 2023

To the Editors of PLoS ONE,

We appreciate your prompt review of our manuscript, “Validation of an omega-3 substrate challenge absorption test as an indicator of global fat lipolysis” (PONE-D-23-01728). Our responses to the Reviewers’ comments are below, indicated by asterisks. 

Reviewer:

Under Abstract 

The section is well written, conclusion is very much clear and concise

* Thank you.

Under Introduction 

Comments

Overall the introduction is good and informative. I would only suggest to put some lines about the functional role of the omega-3 substrate challenge absorption test in this section rather than focusing more on the critical aspect of the coefficient of fat absorption (CFA) test. 

* We have deleted some of the discussion of CFA and added a sentence to highlight the functional aspects of the omega-3 SACT.

Under Methodology

The methodology employed is robust, however information about following points need to be addressed:

• The methodology portion should be limited to the information about use of methods. You have added the background of exocrine pancreatic insufficient (EPI) pig model which is not needed here. 

* We deleted some of the more specific description of the EPI pig model.

• Give information about the following points:

o Is the sample size sufficient? 

* The sample size was based on previous experiments using the EPI pig model and was estimated using G*Power software, version 3.1.9.7 for ANOVA with repeated measurements and within-between interactions, at α = 0:05 with 95% power, assuming f (effect size) = 1.2, for five study groups and five measurements, taking into consideration the correction for repeated measurements (0.5). The calculation yielded the total sample size of 10 animals and the 12 animals were taken considering the possible risk of EPI surgery failure (1, 2, 3). This has been added to the Methods section.

o How many replications were performed? 

* Five replications were performed.

o Time of blood sampling?

* As described in the Methods section in the last paragraph of the section sub-titled “Exocrine pancreatic insufficient (EPI) pig model”, the timing of blood sampling was 0, 1, 2, 4, 6, 8, 12, and 24 hours. We added that the first (0) sampling was performed at 8:00 am local time.

o Sampling procedure, use of syringes and anticoagulant use is missing, time gap between blood sampling and analysis. 

* Blood samples were taken from the jugular vein catheter to 5 ml syringes (BD Syringe™, SKU 309646, BD Switzerland Sarl, 1262 Eysins, Vaud. Switzerland) and then transferred to BD Vacutainer® EDTA Tubes (SKU 366643, BD Switzerland Sarl, 1262 Eysins, Vaud. Switzerland). The collected blood was centrifuged at 4,000 rpm at 4°C for 10 min. Plasma was separated and stored at -80C. Plasma was analyzed within one month after study completion. This has been added to the Methods section.

o Features of the column use for fatty acid determination in GCMS.

* Fatty acid analysis was carried out using a GC2010 Gas Chromatopgraph (Shimadzu) equipped with a SP2560 100-m fused silica capillary column. We have added this to the Methods section and provided a reference with the complete methodology (4).

Under statistical analysis 

Robust 

* We appreciate your external review.

Under results and discussion

Both the results are very informative. Well written. 

No major changes need in this section. 

• You should further relate your study with other such work being carried over recently. 

* We have added more description of the two relevant studies cited and how they relate to this work in the Discussion. We changed the order in which they are presented to improve the flow of the paragraph.

• Try to make this portion more coherent and improve the figure quality

* We have expanded the sub-section title in the Results to make the work more coherent.

* Revised Figures have been re-submitted as TIFF files.

All the authors appreciate the Reviewers’ input and agree with these revisions to this submission. We hope you will accept these changes and consider this revised manuscript for publication in PLoS ONE.

Sincerely,

Drucy Borowitz, MD

Corresponding author

1. Donaldson J, Fed'kiv O, Pawlowska M, Kowalik S, Erlwanger KH, Westrom B, et al. The effectiveness of enzymatic replacement therapy measured by turbidimetry and the lipaemic index in exocrine pancreatic insufficient young, growing pigs, fed a high-fat diet. Adv Med Sci. 2009;54(1):7-13.

2. Pierzynowska K, Valverde-Piedra J, Szymanczyk S, Prykhod'ko O, Pieszka M, Kardas M, et al. Pancreatic-like enzymes of microbial origin restore growth and normalize lipid absorption in a pig model with exocrine pancreatic insufficiency. Arch Med Sci. 2018;14(2):407-14.

3. Faul F, Erdfelder E, Lang AG, Buchner A. G*Power 3: a flexible statistical power analysis program for the social, behavioral, and biomedical sciences. Behav Res Methods. 2007;39(2):175-91.

4. Harris WS, Pottala JV, Vasan RS, Larson MG, Robins SJ. Changes in erythrocyte membrane trans and marine fatty acids between 1999 and 2006 in older Americans. J Nutr. 2012;142(7):1297-303.

---

## [Editor Report · Decision Letter 1]

5 Apr 2023

Validation of an omega-3 substrate challenge absorption test as an indicator of global fat lipolysis

PONE-D-23-01728R1

Dear Dr. Drucy S. Borowitz

We’re pleased to inform you that your manuscript has been judged scientifically suitable for publication and will be formally accepted for publication once it meets all outstanding technical requirements.

Kind regards,

Ishtiyaq Ahmad, Ph.D

Academic Editor

PLOS ONE
---

## [Editor Report · Acceptance letter]

29 Apr 2023

PONE-D-23-01728R1 

Validation of an omega-3 substrate challenge absorption test as an indicator of global fat lipolysis 

Dear Dr. Borowitz:

I'm pleased to inform you that your manuscript has been deemed suitable for publication in PLOS ONE. Congratulations! Your manuscript is now with our production department. 

Kind regards, 

on behalf of

Dr. Ishtiyaq Ahmad 

Academic Editor

PLOS ONE